# Metabolically healthy obesity, transition to unhealthy metabolic status, and vascular disease in Chinese adults: A cohort study

Meng Gao[1], Jun Lv[1,2,3], Canqing Yu[1], Yu Guo[4], Zheng Bian[4], Ruotong Yang[1], Huaidong Du[5,6], Ling Yang[5,6], Yiping Chen[5,6], Zhongxiao Li[7], Xi Zhang[7], Junshi Chen[8], Lu Qi[9,10], Zhengming Chen[6], Tao Huang[1,2]*, Liming Li[1]*, for the China Kadoorie Biobank (CKB) Collaborative Group[¶]

1 Department of Epidemiology and Biostatistics, School of Public Health, Peking University Health Science Center, Beijing, China, 2 Key Laboratory of Molecular Cardiovascular Sciences (Peking University), Ministry of Education, Beijing, China, 3 Peking University Institute of Environmental Medicine, Beijing, China, 4 Chinese Academy of Medical Sciences, Beijing, China, 5 Medical Research Council Population Health Research Unit at the University of Oxford, Oxford, United Kingdom, 6 Clinical Trial Service Unit & Epidemiological Studies Unit (CTSU), Nuffield Department of Population Health, University of Oxford, Oxford, United Kingdom, 7 Maiji Center for Disease Control and Prevention, Maiji, Gansu, China, 8 China National Center for Food Safety Risk Assessment, Beijing, China, 9 Department of Epidemiology, School of Public Health and Tropical Medicine, Tulane University, New Orleans, Louisiana, United States of America, 10 Department of Nutrition, Harvard T.H. Chan School of Public Health, Boston, Massachusetts, United States of America

[¶]The members of steering committee and collaborative group are listed in the Acknowledgments.
* huangtao@bjmu.edu.cn (TH); lmlee@vip.163.com (LL)

**Data Availability Statement:** Details of how to access China Kadoorie Biobank data and details of the data release schedule are available from www.ckbiobank.org/site/Data+Access

## Abstract

### Background

Metabolically healthy obesity (MHO) and its transition to unhealthy metabolic status have been associated with risk of cardiovascular disease (CVD) in Western populations. However, it is unclear to what extent metabolic health changes over time and whether such transition affects risks of subtypes of CVD in Chinese adults. We aimed to examine the association of metabolic health status and its transition with risks of subtypes of vascular disease across body mass index (BMI) categories.

### Methods and findings

The China Kadoorie Biobank was conducted during 25 June 2004 to 15 July 2008 in 5 urban (Harbin, Qingdao, Suzhou, Liuzhou, and Haikou) and 5 rural (Henan, Gansu, Sichuan, Zhejiang, and Hunan) regions across China. BMI and metabolic health information were collected. We classified participants into BMI categories: normal weight (BMI 18.5–23.9 kg/m²), overweight (BMI 24.0–27.9 kg/m²), and obese (BMI ≥ 28 kg/m²). Metabolic health was defined as meeting less than 2 of the following 4 criteria (elevated waist circumference, hypertension, elevated plasma glucose level, and dyslipidemia). The changes in obesity and metabolic health status were defined from baseline to the second resurvey with combination of overweight and obesity. Among the 458,246 participants with complete information and no history of CVD and cancer, the mean age at baseline was 50.9 (SD 10.4) years, and

**Funding:** LL and JL received grants (2016YFC0900500, 2016YFC0900501, 2016YFC0900504, 2016YFC1303904) from the National Key R&D Program of China. TH received grants (2019YFC2003400) from the National Key R&D Program of China. ZMC received grants from the UK Wellcome Trust (212946/Z/18/Z, 202922/Z/16/Z, 104085/Z/14/Z, 088158/Z/09/Z); LL and JL received grants from National Natural Science Foundation of China (91846303, 91843302, 81390540, 81390541, 81390544), and Chinese Ministry of Science and Technology (2011BAI09B01). The funders had no role in study design, data collection and analysis, decision to publish, or preparation of the manuscript.

**Competing interests:** The authors have declared that no competing interests exist.

**Abbreviations:** BMI, body mass index; BRICS, Brazil, Russia, India, China, and South Africa; CDC, Center for Disease Control and Prevention; CI, confidence interval; CKB, China Kadoorie Biobank; CVD, cardiovascular disease; HR, hazard ratio; IHD, ischemic heart disease; MCE, major coronary event; MHN, metabolically healthy normal weight; MHO, metabolically healthy obesity; MHOO, metabolically healthy overweight or obesity; MUN, metabolically unhealthy normal weight; MUO, metabolically unhealthy obesity; MUOO, metabolically unhealthy overweight or obesity; MVE, major vascular events; RPG, random plasma glucose.

40.8% were men, and 29.0% were current smokers. During a median 10.0 years of follow-up, 52,251 major vascular events (MVEs), including 7,326 major coronary events (MCEs), 37,992 ischemic heart disease (IHD), and 42,951 strokes were recorded. Compared with metabolically healthy normal weight (MHN), baseline MHO was associated with higher hazard ratios (HRs) for all types of CVD; however, almost 40% of those participants transitioned to metabolically unhealthy status. Stable metabolically unhealthy overweight or obesity (MUOO) (HR 2.22, 95% confidence interval [CI] 2.00–2.47, $p < 0.001$) and transition from metabolically healthy to unhealthy status (HR 1.53, 1.34–1.75, $p < 0.001$) were associated with higher risk for MVE, compared with stable healthy normal weight. Similar patterns were observed for MCE, IHD, and stroke. Limitations of the analysis included lack of measurement of lipid components, fasting plasma glucose, and visceral fat, and there might be possible misclassification.

## Conclusions

Among Chinese adults, MHO individuals have increased risks of MVE. Obesity remains a risk factor for CVD independent of major metabolic factors. Our data further suggest that metabolic health is a transient state for a large proportion of Chinese adults, with the highest vascular risk among those remained MUOO.

## Author summary

### Why was this study done?

- Obesity affects more than 10% of the Chinese adults and may cause metabolic disorder and cardiovascular disease.

- People with obesity have variability in metabolic factors, and a subset of individuals with obesity do not develop metabolic disorders. There is limited prospective evidence on the combined association of obesity and metabolic health status and their transition over time with incident cardiovascular disease.

### What did the researchers do and find?

- We conducted a cohort study using data of 458,246 participants from 5 urban and 5 rural areas across China during 2004–2008 and then tracked their health until 31 December 2016.

- Individuals with obesity and metabolic health status had a significantly higher risk of developing major vascular events.

- About 40% of overweight or obese participants with metabolic health status developed unhealthy status. These individuals also had a significantly higher risk of incident major vascular events, and the risk is lower than those with stable overweight or obesity and metabolic unhealthy status.

**What do these findings mean?**

- The present study supports that obesity with metabolic health status was a relatively harmful condition for cardiovascular disease, and obesity remains a major risk factor for cardiovascular disease independent of common metabolic disorders.

- Metabolic health is a transient state for a large proportion of Chinese adults. Our findings highlighted the importance of maintaining metabolic health across all BMI groups in early prevention of vascular events.

## Introduction

Cardiovascular disease (CVD) is a leading cause of death and disability worldwide and contributes to more than 17 million deaths annually [1], especially 8.4 million CVD deaths across Brazil, Russia, India, China, and South Africa (BRICS) in 2016 [2]. Obesity and its related metabolic disorders have been major risk factors for CVD globally, including in China [3–5]. However, people with obesity have variability in metabolic factors. It has been reported that a subset of individuals with obesity do not develop metabolic disorders [6, 7] and are described as having metabolically healthy obesity (MHO), though most Western studies suggested MHO is not an absolute healthy status for diabetes and CVD.

Previous cohort studies have shown that MHO phenotype is associated with a higher risk of CVD compared with individuals with metabolically healthy normal weight (MHN) [8–12], although inconsistent results have also been reported [13–16]. Furthermore, a meta-analysis demonstrated that such increased CVD risk of MHO individuals is considerably lower than that of individuals with metabolically unhealthy obesity (MUO) [17]. However, these estimates were mostly from Western populations, with little evidence from China [18, 19], where adiposity distribution, risk of obesity, lifestyle, and disease patterns differ substantially from those in Western populations [20–22]. Importantly, the Western cohort studies demonstrated that metabolic health changed over time across body mass index (BMI) categories and was associated with cardiovascular risk [9, 23]. However, it remains unclear how metabolic factors change over a long time across BMI groups and how such dynamic metabolic changes affects vascular risk among Chinese adults. Studies assessing the cardiovascular hazards of dynamic metabolic changes over time in low- and middle-income countries, including China, are of both public health and clinical significance and are needed to inform disease prevention strategies.

Therefore, our study aimed to examine the associations of BMI categories and metabolic health status and their transition over time with CVD, including major vascular events (MVEs), major coronary events (MCEs), ischemic heart disease (IHD), and stroke in the China Kadoorie Biobank (CKB) study, an ongoing prospective cohort of about 0.5 million adults.

## Methods

This study is reported as per the Strengthening the Reporting of Observational Studies in Epidemiology (STROBE) guideline (S1 STROBE Checklist). For the current study, the analysis plan was drafted in December 2018 (S1 Text).

## Study population

The CKB cohort was established in 10 (5 urban and 5 rural) regions geographically spread across China. The study design, methods, and participants have been described in detail previously [24, 25]. Briefly, a total of 512,715 participants aged 30 to 79 years old were enrolled in the study during 25 June 2004 to 15 July 2008, and the participation rate was about 30%. Two periodic resurveys were conducted in 2008 and during 4 August 2013 to 18 September 2014, on approximately 5% of randomly chosen surviving participants using administrative unit as the basic sampling unit. The second resurvey was a representative sample of baseline sample. At baseline and subsequent resurveys, information on sociodemographic characteristics, lifestyles, medical history, and physical measurements were collected by trained staff (S2 Text). In this study, we excluded participants with a prior history of coronary heart disease ($n$ = 15,472), stroke ($n$ = 8,884), or cancer ($n$ = 2,578), as well as individuals with missing values of BMI ($n$ = 2) or plasma glucose ($n$ = 8,160). Besides, underweight participants (BMI < 18.5 kg/m$^2$) were excluded ($n$ = 22,361). Finally, a total of 458,246 participants (187,168 men and 271,078 women) were included in the analysis.

The Ethical Review Committee of the Chinese Center for Disease Control and Prevention (Beijing, China) and the Oxford Tropical Research Ethics Committee, University of Oxford (UK) approved the study.

## Measurement of adiposity and metabolic factors

Standing height was measured to the nearest 0.1 cm with the participant standing erect in bare feet. Weight was measured to the nearest 0.1 kg using the TBF-300 body composition analyzer (Tanita Inc, Tokyo, Japan) and the estimated weight of clothing subtracted (summer 0.5 kg; spring/autumn 1.0 kg; winter 2.0–2.5 kg). BMI was calculated as weight in kilograms dividing by the square of height in meters. Waist circumference was measured to the nearest 0.1 cm using a soft nonstretchable tape at the midpoint between the lowest rib margin and the iliac crest. Blood pressure was measured at least twice using a UA-779 digital monitor, and the mean of the 2 measurements qualified was used in the analysis. A nonfasting venous blood sample was collected from participants, and the time passed since participants last ate was recorded. Immediately, on-site testing of plasma glucose level was undertaken using the Sure-Step Plus meter (LifeScan, Milpitas, CA, USA). Participants with a glucose level ≥7.8 mmol/L and <11.1 mmol/L were invited to return the following day for fasting plasma glucose testing. In the second resurvey, plasma triglycerides (TG), low-density lipoprotein cholesterol (LDL-C), and high-density lipoprotein cholesterol (HDL-C) were measured using Mission Cholesterol Monitoring System (ACON, Hangzhou, China).

## Assessment of BMI categories and metabolic health status and their transition

We classified participants into BMI categories based on Chinese guideline [26]: normal weight (BMI 18.5–23.9 kg/m$^2$), overweight (BMI 24.0–27.9 kg/m$^2$), and obese (BMI ≥28 kg/m$^2$). We defined metabolic health based on a modified definition of the metabolic syndrome, as described by the joint statement in 2009 [27]. In baseline, participants who met <2 of the following 4 criteria were considered metabolically healthy: (1) waist circumference ≥90 cm for men and ≥85 cm for women; (2) systolic blood pressure ≥130 mmHg or diastolic blood pressure ≥85 mmHg or self-reported hypertension or using antihypertensive drugs; (3) fasting plasma glucose (FPG) ≥5.6 mmol/L or random plasma glucose (RPG) ≥11.1 mmol/L or self-reported diabetes; (4) using lipid-lowing drugs. Because the dyslipidemia was assessed by self-

reported lipid-lowing drug use at baseline, the main analysis was repeated in participants recruited in the second resurvey with TG and HDL-C data, to reduce the bias from the mis-classification of metabolic health status at baseline. In the second resurvey, participants who met <3 of the following 5 criteria were considered metabolically healthy: (1) waist circumfer-ence ≥90 cm for men and ≥85 cm for women; (2) systolic blood pressure ≥130 mmHg or dia-stolic blood pressure ≥85 mmHg or self-reported hypertension or using antihypertensive drugs; (3) FPG ≥5.6 mmol/L or RPG ≥11.1 mmol/L or self-reported diabetes; (4) reduced plasma HDL-C (<1.0 mmol/L for men and <1.3 mmol/L for women) or using lipid-lowing drugs; and (5) elevated plasma TG (≥1.7 mmol/L) or using lipid-lowing drugs.

Based on the combination of BMI categories and metabolic health status, participants were then categorized into 6 groups: MHN; metabolically healthy overweight (MHOW); MHO; metabolically unhealthy normal weight (MUN); metabolically unhealthy overweight (MUOW); and MUO. In addition, we defined transitions (MHN throughout, MHN to meta-bolically healthy overweight or obesity [MHOO], MHOO throughout, MHOO to metaboli-cally unhealthy overweight or obesity [MUOO]) from baseline to the second resurvey, with combination of overweight and obesity for small sample size.

## Ascertainment of outcomes

Incident outcome cases since the participants' enrollment into the study at baseline were iden-tified by using the linkage with local disease and death registries, checked against the national health insurance system, or ascertained through active follow-up [25]. Fewer than 1% of par-ticipants (n = 4,749) were lost to follow-up before the end of the study. Vital status and cause of death were monitored regularly through official residential records and death certificates reported to the regional Center for Disease Control and Prevention (CDC) in 10 regions. Information on disease incidence for IHD and stroke is also being collected through linkage with established disease registries in 8 out of the 10 study regions. Linkage to local health insur-ance databases had already been achieved for 91% of the participants by 1 January 2011 [25], and active follow-up was conducted annually for participants who were not linked to their local health insurance database. Besides, the medical records of cases were retrieved, and the diagnosis was adjudicated centrally by qualified cardiovascular specialists blinded to study assay. By 31 December 2013, of 20,154 incident IHD cases and 20,154 incident stroke cases reported since baseline and from patients whose medical records have been retrieved, the diag-nosis was confirmed in 83% of IHD cases and in 91% of stroke cases. All cases were coded using the 10th Revision of International Classification of Diseases (ICD-10) by trained staff blinded to baseline information. The primary outcomes were incident MVE (including vascu-lar [codes I00 to I99] death, nonfatal MI [I21 to I23] and nonfatal stroke [I60, I61, I63, and I64]), MCE (including IHD [I20 to I25] death and nonfatal myocardial infarction [I21 to I23]), IHD (I20 to I25), and stroke (I60, I61, I63, and I64).

## Statistical analysis

Age-, sex-, and study region–adjusted characteristics of the study population were described as percentages or means (SDs), where logistic regression and multiple linear regression were implemented for categorical variables and continuous variables, respectively (age, sex, and urban region themselves were not adjusted). Person-time of follow-up was calculated from baseline or the second resurvey until a report of cardiovascular disease event, death, loss to fol-low-up, or the end of follow-up (31 December 2016), whichever came first. Cox proportional hazard models were used with age as the time scale to estimate the hazard ratios (HRs) for inci-dent CVDs by BMI-metabolic health status. The corresponding 95% confidence intervals (CIs)

of HRs were calculated by use of the floating absolute risk method [28] to enable comparisons between any 2 categories. The proportional hazard assumption was examined by Schoenfeld residuals. The multivariate model was adjusted for age (5 years); study region (10 regions); sex (men or women); education (middle school or less, high school or above); household income (<20,000, or ≥20,000 yuan/year); marital status (married, or others); smoking (current regular smoker, or not current regular smoker); alcohol use (weekly alcohol consumer, or nonweekly consumer); intakes of red meat, fresh fruits, and vegetables (daily, 4–6 days/week, 1–3 days/week, monthly, or never/rarely); physical activity (based on tertiles, MET-hour/day); and family history of heart attack or stroke (presence or absence). We examined the joint effects of BMI-metabolic health status with age (4 groups) or sex (men or women) to estimate the age- and sex-specific associations. We also tested the nonlinear association between the aforementioned risk factors of metabolic health (blood pressure, waist circumference, RPG) and MVE using a restricted cubic spline function. We examined the association between the number of metabolic disorders participants met and the development of MVE. In addition, HRs and corresponding 95% CIs were also calculated for incident CVD types by changes of BMI-metabolic health status with the same model, and the reference group is participants with stable MHN.

To examine the robustness of our results, we performed several sensitivity analyses: excluding cases occurring in the first 2 years of follow-up; excluding ever smokers; additionally adjusting for the amount of cigarettes consumed per day (1–14, 15–24, ≥25) and the amount of alcohol consumed (<15, 15–29, 30–59, ≥60 g/day); additionally adjusting for systolic blood pressure (by quintile, <114, 114–123.4, 123.5–132.4, 132.5–146.4, ≥146.5 mm Hg) and RPG (by quintile, <4.8, 4.8–5.2, 5.3–5.7, 5.8–6.6, ≥6.7 mmol/L) to report whether the associations were caused by difference of blood pressure and RPG; using waist–hip ratio (≥0.90 for men and ≥0.85 for women) instead of the waist circumference criterion defined metabolic health status; using waist–height ratio (≥0.50) instead of the waist circumference criterion; using continuous variables (intakes of red meat, fresh fruits, and vegetables [day/week], and physical activity [MET-hour/day]) instead of factor variables; using an alternative definition of metabolic health that none of elevated blood pressure, elevated plasma glucose, and lipid-lowing drugs use, excluding waist circumference criterion for high correlation with BMI. All statistical analyses were performed with Stata version 15.0 (StataCorp) and SAS version 9.4 (SAS Institute Inc., Cary, NC, USA). All p-values were 2-sided, and statistical significance was defined as $p < 0.05$.

### Patient and public involvement

Patients were not involved in the present study. The results of the main study were presented to study participants at the website of the CKB study (http://www.ckbiobank.org/site/) and by newsletters annually.

## Results

### Baseline characteristics of participants by BMI-metabolic health status

Among the 458,246 participants, the mean (SD) age at baseline was 50.9 (10.4) years, and 40.8% were men. At baseline, 22.4% (n = 102,710) of the participants were metabolically unhealthy, and 10.7% (n = 49,168) had obesity. The MHO phenotype accounted for 3.3% (n = 15,044) of the total population and 30.6% of the obese population. The prevalence of MHO was higher in women than in men and in young people than in old people at baseline and the second resurvey (Fig 1). Age-specific and sex-specific prevalences of MHO were presented in S1 and S2 Figs. Baseline and the second resurvey characteristics of the study population according to BMI-metabolic health status are shown in Tables 1 and S1. Metabolically healthy individuals were more likely to be younger and more physically active across all BMI

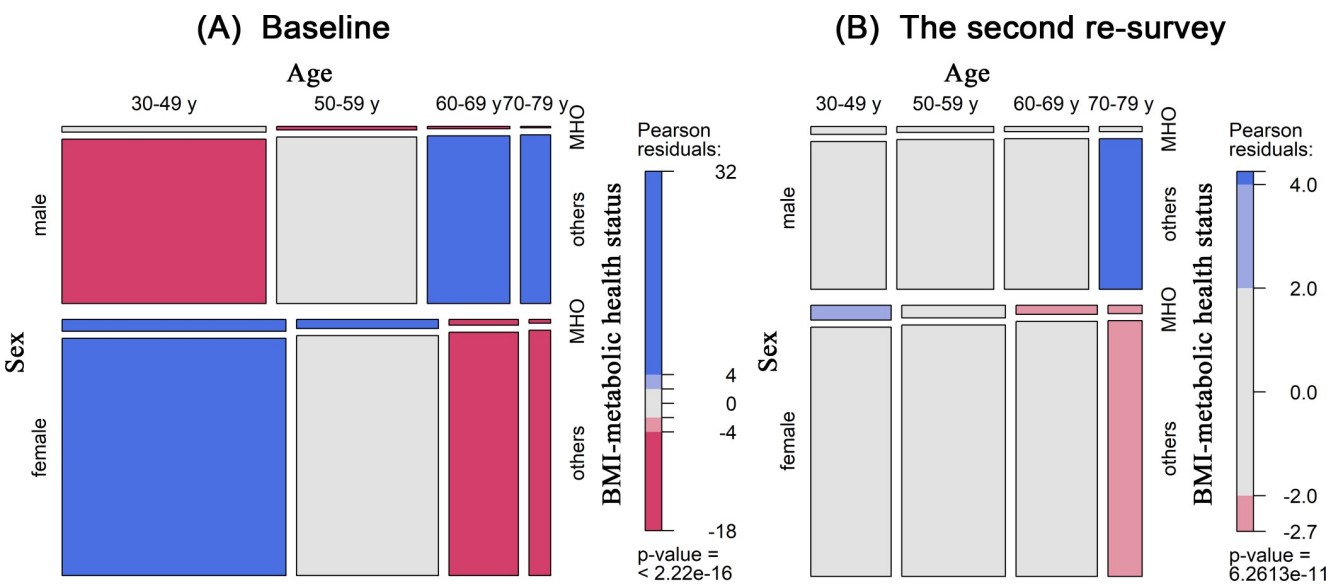

**Fig 1. The prevalences of MHO by age and sex.** (**A**) Baseline and (**B**) the second resurvey. BMI, body mass index; MHO, metabolically healthy obesity.

categories. MHO individuals were more likely to be younger, female, and nonweekly drinkers compared with MUO individuals.

## Associations of BMI-metabolic health status with cardiovascular disease

During a median follow-up of 10.0 years, there were 52,251 MVEs, including 7,326 MCEs, 37,992 IHDs, and 42,951 strokes. MHO individuals had an 8% higher risk of developing MVEs (HR, 1.08; 95% CI 1.02–1.14; $p$ = 0.009) compared with MHN individuals, and the corresponding HR for MUO was 1.67 (95% CI 1.63–1.72, $p$ < 0.001). For specific types of CVD, the adjusted HRs for the MHO and MUO individuals were 1.05 (0.89–1.25, $p$ = 0.557) and 1.92 (1.80–2.05, $p$ < 0.001) for MCE, 1.34 (1.27–1.42, $p$ < 0.001) and 1.79 (1.74–1.84, $p$ < 0.001) for IHD, and 1.11 (1.05–1.18, $p$ = 0.001) and 1.71 (1.66–1.76, $p$ < 0.001) for stroke, respectively (Table 2 and Fig 2). In the second resurvey, the results were similar. MHO individuals had higher risk of IHD; however, the CIs for MVE, MCE, and stroke were large. MUO individuals had higher risk of MVE, stroke, and IHD. We found systolic blood pressure, diastolic blood pressure, and RPG had nonlinear association with major vascular disease ($p$ < 0.05) and no evidence of nonlinear association between waist circumference and major vascular disease ($p$ = 0.088) (S4 Fig). We found higher risk of major vascular disease with the increase of number of metabolic disorders (S5 Fig).

The age- and sex-specific associations are presented in Figs S3 and 3. MUO individuals at 70 to 79 years had the highest risk of developing MVE among 24 groups classified by age and BMI-metabolic health status (HR 13.86 [12.97–14.80], $p$ < 0.001), with MHN individuals at age 30 to 49 years as the reference group. In each age group, MHO individuals had a higher risk of developing MVE compared with MHN individuals, and the risk was considerably higher in metabolically unhealthy ones, whereas the corresponding associations were attenuated in older people. Similar associations were observed for MCE, IHD, and stroke (S3 Fig). Men had a higher risk of each subtype of CVD than women, except for IHD, for which women had a higher risk. The HRs for BMI-metabolic health status and types of CVD were similar between men and women (Fig 3).

**Table 1. Baseline characteristics of the study population.**

| Characteristics | All | MHN | MHOW | MHO | MUN | MUOW | MUO |
|---|---|---|---|---|---|---|---|
| No. of participants | 458,246 | 232,975 | 107,517 | 15,044 | 18,428 | 50,158 | 34,124 |
| **Demographic factors** | | | | | | | |
| Age (y) | 50.9 (10.4) | 50.2 (10.6) | 49.2 (9.6) | 47.7 (9.1) | 57.5 (9.9) | 54.9 (10.0) | 53.0 (10.0) |
| Male (%) | 40.8 | 43.0 | 39.8 | 29.9 | 35.3 | 41.3 | 36.7 |
| Urban (%) | 44.3 | 38.0 | 49.2 | 53.4 | 44.2 | 52.3 | 56.3 |
| **Socioeconomic factors (%)** | | | | | | | |
| Middle school and above | 50.1 | 50.5 | 51.0 | 49.4 | 49.5 | 49.1 | 46.7 |
| Household income ≥20,000 yuan/year | 43.2 | 41.7 | 44.0 | 43.3 | 44.3 | 46.6 | 44.5 |
| Married | 91.1 | 90.4 | 92.2 | 92.4 | 90.5 | 91.8 | 91.6 |
| **Lifestyle factors** | | | | | | | |
| Current smoker (%) | 29.0 | 30.6 | 26.9 | 27.0 | 29.0 | 27.5 | 26.7 |
| Current smoker-male (%) | 67.3 | 71.1 | 62.6 | 62.5 | 68.0 | 63.9 | 61.1 |
| Current smoker-female (%) | 2.5 | 2.8 | 2.2 | 2.3 | 2.5 | 2.2 | 2.4 |
| Weekly drinker (%) | 15.1 | 15.1 | 14.6 | 13.9 | 16.6 | 16.0 | 15.3 |
| Weekly drinker, male (%) | 34.2 | 34.1 | 32.8 | 30.9 | 38.3 | 36.5 | 34.7 |
| Weekly drinker, female (%) | 2.0 | 2.1 | 2.0 | 2.0 | 1.7 | 1.7 | 1.8 |
| Physical activity (MET-h/d) | 21.5 (13.9) | 22.2 (14.1) | 21.5 (13.8) | 20.6 (13.0) | 20.8 (13.8) | 20.4 (13.1) | 19.6 (12.7) |
| Meat intake (day/week) | 3.7 (2.5) | 3.7 (2.5) | 3.8 (2.6) | 3.8 (2.6) | 3.7 (2.5) | 3.8 (2.6) | 3.8 (2.6) |
| Vegetable intake (day/week) | 6.8 (0.8) | 6.8 (0.8) | 6.9 (0.7) | 6.8 (0.7) | 6.8 (0.8) | 6.9 (0.7) | 6.9 (0.6) |
| Fruit intake (day/week) | 2.6 (2.5) | 2.6 (2.4) | 2.7 (2.5) | 2.8 (2.7) | 2.3 (2.4) | 2.5 (2.6) | 2.6 (2.7) |
| **Physical measurements** | | | | | | | |
| BMI (kg/m$^2$) | 23.9 (3.2) | 21.6 (1.5) | 25.4 (1.1) | 29.4 (1.6) | 22.2 (1.4) | 26.1 (1.1) | 30.1 (2.1) |
| WC (cm) | 80.8 (9.3) | 74.8 (6.0) | 83.3 (5.5) | 91.5 (7.4) | 79.5 (7.8) | 89.6 (5.4) | 96.3 (6.8) |
| Waist–hip ratio | 0.9 (0.1) | 0.9 (0.1) | 0.9 (0.1) | 0.9 (0.1) | 0.9 (0.1) | 0.9 (0.1) | 1.0 (0.1) |
| SBP (mmHg) | 131.0 (21.0) | 125.8 (19.6) | 129.6 (19.0) | 126.4 (15.1) | 143.4 (18.8) | 144.1 (19.1) | 146.9 (19.5) |
| DBP (mmHg) | 77.9 (11.1) | 75.2 (10.4) | 77.6 (10.4) | 76.5 (8.8) | 82.7 (10.5) | 84.3 (10.5) | 85.8 (10.9) |
| **Self-reported conditions (%)** | | | | | | | |
| Elevated WC | 24.6 | 1.2 | 18.2 | 67.8 | 27.5 | 83.8 | 98.4 |
| Elevated BP | 50.1 | 37.0 | 42.6 | 22.2 | 96.7 | 92.9 | 94.4 |
| Elevated plasma glucose | 12.9 | 5.1 | 4.0 | 1.5 | 76.4 | 35.2 | 24.8 |
| **Family medical history (%)** | | | | | | | |
| Stroke | 17.9 | 17.1 | 18.1 | 16.9 | 18.9 | 19.8 | 20.0 |
| Heart attack | 3.2 | 3.1 | 3.2 | 3.0 | 3.4 | 3.4 | 3.5 |

Baseline characteristics of the study population were described adjusted for age, sex and region except for number of participants, age, sex, and urban region.

BMI, body mass index; BP, blood pressure; DBP, diastolic blood pressure; MET-h/d, metabolic equivalents of task per hours per day; MHN, metabolically healthy normal weight; MHO, metabolically healthy obesity; MHOW, metabolically healthy overweight; MUN, metabolically unhealthy normal weight; MUO, metabolically unhealthy obesity; MUOW, metabolically unhealthy overweight; SBP, systolic blood pressure; WC, waist circumference.

In the sensitivity analysis, the associations of MHO individuals with MVEs were not materially altered when further adjusted for systolic blood pressure and RPG, whereas the associations of metabolically unhealthy individuals obviously attenuated. Other results did not change by excluding cases within the first 2 years, further adjustment for other potential confounders, excluding ever smokers, using waist–hip ratio or waist–height ratio instead of waist circumference, using continuous variables instead of factor variables, and using the alternative definition excluding waist circumference criterion (S3 and S4 Tables).

**Table 2. Adjusted HRs for vascular diseases by BMI-metabolic health status at baseline.**

| | MHN | MHOW | MHO | MUN | MUOW | MUO |
|---|---|---|---|---|---|---|
| **Major vascular events** | | | | | | |
| Cases | 22,208 | 9,756 | 1,222 | 3,673 | 9,308 | 6,084 |
| Person-years | 2,263,506 | 1,058,246 | 149,650 | 163,919 | 457,010 | 317,883 |
| HR, model 1 | 1.00 (0.99–1.01) | 1.04 (1.02–1.06) | 1.08 (1.02–1.15) | 1.50 (1.45–1.55) | 1.58 (1.54–1.61) | 1.69 (1.65–1.74) |
| HR, model 2 | 1.00 (0.99–1.01) | 1.06 (1.04–1.08) | 1.08 (1.02–1.14) | 1.54 (1.49–1.60) | 1.58 (1.55–1.62) | 1.67 (1.63–1.72) |
| **Major coronary events** | | | | | | |
| Cases | 3,052 | 1,178 | 134 | 616 | 1,424 | 922 |
| Person-years | 2,320,018 | 1,087,933 | 153,411 | 173,541 | 484,710 | 336,148 |
| HR, model 1 | 1.00 (0.96–1.04) | 0.98 (0.92–1.04) | 1.02 (0.86–1.21) | 1.69 (1.56–1.83) | 1.66 (1.58–1.75) | 1.88 (1.76–2.01) |
| HR, model 2 | 1.00 (0.96–1.04) | 1.02 (0.96–1.08) | 1.05 (0.89–1.25) | 1.78 (1.65–1.93) | 1.73 (1.64–1.82) | 1.92 (1.80–2.05) |
| **Ischemic heart disease** | | | | | | |
| Cases | 15,342 | 7,527 | 1,194 | 2,353 | 6,751 | 4,825 |
| Person-years | 2,269,151 | 1,061,802 | 148,993 | 165,851 | 461,504 | 319,245 |
| HR, model 1 | 1.00 (0.98–1.02) | 1.10 (1.08–1.13) | 1.37 (1.29–1.45) | 1.39 (1.33–1.44) | 1.61 (1.57–1.65) | 1.82 (1.77–1.87) |
| HR, model 2 | 1.00 (0.98–1.02) | 1.10 (1.07–1.12) | 1.34 (1.27–1.42) | 1.41 (1.35–1.47) | 1.60 (1.56–1.64) | 1.79 (1.74–1.84) |
| **Stroke** | | | | | | |
| Cases | 17,697 | 8,352 | 1,064 | 2,905 | 7,815 | 5,118 |
| Person-years | 2,267,260 | 1,060,278 | 149,906 | 164,555 | 458,868 | 319,108 |
| HR, model 1 | 1.00 (0.98–1.02) | 1.09 (1.07–1.12) | 1.13 (1.06–1.20) | 1.50 (1.45–1.56) | 1.65 (1.61–1.69) | 1.74 (1.70–1.79) |
| HR, model 2 | 1.00 (0.98–1.02) | 1.10 (1.07–1.12) | 1.11 (1.05–1.18) | 1.54 (1.48–1.60) | 1.64 (1.61–1.68) | 1.71 (1.66–1.76) |

Multivariable models were adjusted for model 1: study region, age (5 years) and sex (men or women) and model 2: study region, age (5 years), sex (men or women), education (primary school or lower, middle school or higher), household income (<20,000 yuan/year, or ≥20,000 yuan/year), marital status (married, others), smoking status (current regular smoker, not current regular smoker), alcohol use(weekly drinker, not weekly drinker), intakes of red meat, fresh fruits and vegetables (daily, 4–6 days/week, 1–3 days/week, monthly, or never/rarely), family history of heart attack or stroke (presence or absence), and physical activity (3 groups).

BMI, body mass index; HR, hazard ratio; MHN, metabolically healthy normal weight; MHO, metabolically healthy obesity; MHOW, metabolically healthy overweight; MUN, metabolically unhealthy normal weight; MUO, metabolically unhealthy obesity; MUOW, metabolically unhealthy overweight.

## Transition of metabolic health status and its association with vascular risk

Furthermore, we examined the changes in metabolic health status across all BMI groups during follow-up. Among participants with MHN at baseline, only 15.17% converted to MHOO, and 67.53% were unconverted in resurvey. Among participants with MHOO, 39.66% converted to MUOO, and 48.21% were unconverted, whereas the majority (67.53%) of MUOO were unconverted throughout the follow-up (Table 3).

The cumulative incidence of MVE for participants with stable MUOO (2.22, 2.00–2.47, $p < 0.001$) was the highest among all groups (Fig 4), with much higher HR than that for IHD (1.92, 1.71–2.16, $p < 0.001$) but similar with that for stroke (2.21, 1.98–2.47, $p < 0.001$). The incidence for participants who changed from metabolic health to MUOO (1.53, 1.34–1.75, $p < 0.001$) was between participants with stable MHOO (1.10, 0.95–1.27, $p = 0.207$) and with those who were stable MUOO (2.22, 2.00–2.47, $p < 0.001$). The risk for participants who changed from MHN to overweight or obesity was not significant (1.16, 0.95–1.42, $p = 0.140$) but was substantially lower than for participants who became from MHOO to MUOO and who stayed MUOO during follow-up (Fig 4). In addition, we observed a similar pattern for MCE, IHD, and stroke.

## Discussion

Our findings show that in Chinese adults, MHO individuals had an 8% higher risk of developing MVE, 34% higher risk of IHD, and 11% higher risk of stroke, with no association found

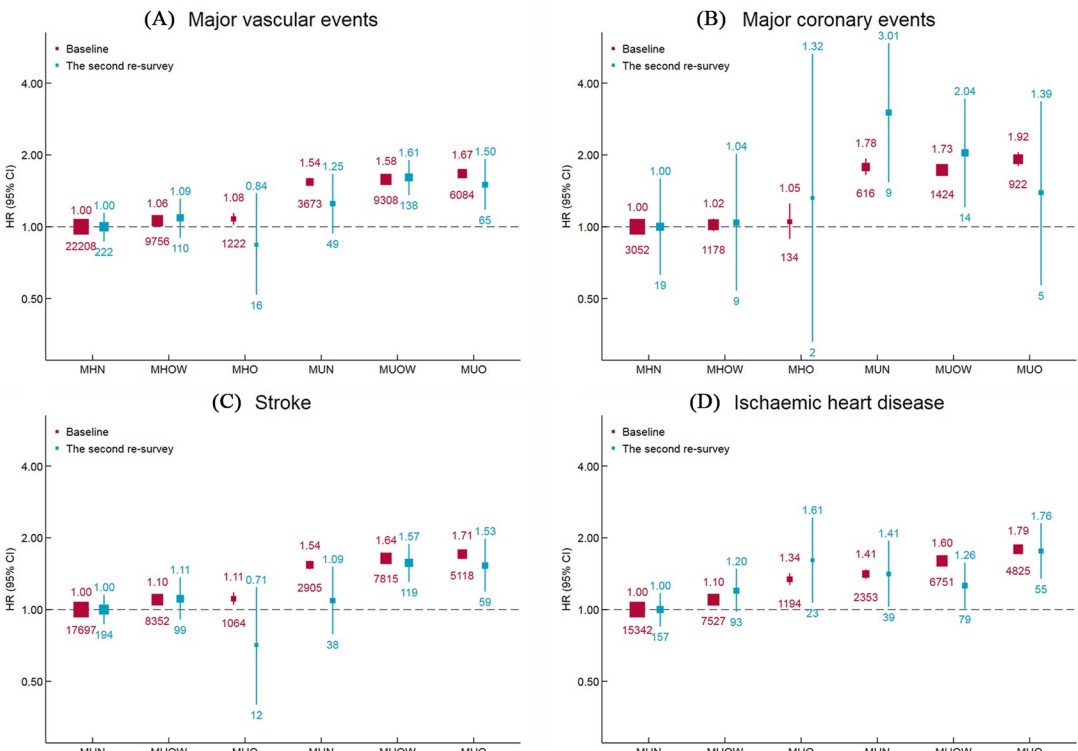

**Fig 2. Adjusted HRs for cardiovascular diseases subtypes by BMI-metabolic health status.** Values shown are the HR (95% CI) for **(A)** major vascular events, **(B)** major coronary events, **(C)** stroke, and **(D)** ischemic heart disease, by BMI-metabolic health status. HRs are adjusted for age, study region, sex, education, household income, marital status, smoking, alcohol use, red meat intake, fresh fruits intake, fresh vegetables intake, physical activity, and family history of heart attack or stroke. The values under the squares indicate number of cases in each category. The vertical lines indicate 95% CIs. The size of the squares is proportional to the inverse variance of each effect size. BMI, body mass index; CI, confidence interval; HR, hazard ratio; MHN, metabolically healthy normal weight; MHO, metabolically healthy obesity; MHOW, metabolically healthy overweight; MUN, metabolically unhealthy normal weight; MUO, metabolically unhealthy obesity; MUOW, metabolically unhealthy overweight.

with MCE. Our data show that the risk of all types of vascular disease in metabolically unhealthy individuals was much higher across BMI categories. These associations were similar in male and female but slightly attenuated in older people. These results supported the notion that obesity remains an independent risk factor for vascular disease. Our data further suggest that metabolic health changes over time across BMI categories. Particularly, stable unhealthy obesity substantially increased the risks of all types of vascular disease, with much higher risk than did the transition from healthy to unhealthy obesity.

Our present results were consistent with those of previous cohort studies indicating that MHO individuals were at an increased risk of incident CVD. A meta-analysis of 13 large prospective studies in Western populations reported that MHO individuals were at a 45% increased cardiovascular risk [17], whereas individuals with both MUN and obesity were at much higher risk than that of individuals with MHO. However, no Asian studies were included in this meta-analysis. More recently, the Beijing Cohort Study [18] and the China Health and Retirement Longitudinal Study [19], including 9,393 and 7,849 Chinese adults, showed that MHO individuals had a higher risk of CVD (HR 1.91 [1.13–3.24] and 1.33 [1.19–1.49]) than MHN ones. However, both studies have a smaller sample size and shorter follow-up time than ours, with very few cases in subgroups. Our findings from the largest prospective cohort study of approximately 500,000 Chinese participants show that MHO was associated

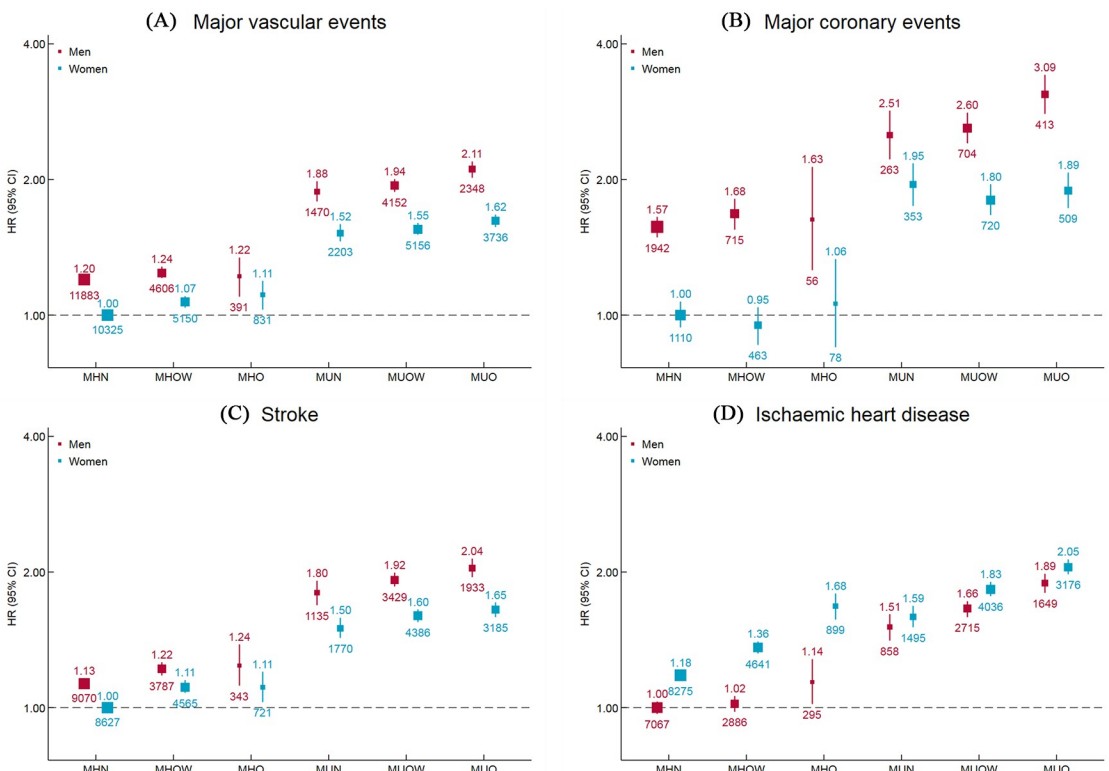

**Fig 3. Adjusted HRs for cardiovascular diseases subtypes by BMI-metabolic health status and sex.** Values shown are the HR (95% CI) for **(A)** major vascular events, **(B)** major coronary events, **(C)** stroke, and **(D)** ischemic heart disease, by BMI-metabolic health status and sex. HRs are adjusted for age, study region, education, household income, marital status, smoking, alcohol use, red meat intake, fresh fruits intake, fresh vegetables intake, physical activity, and family history of heart attack or stroke. The vertical lines indicate 95% CIs. The values above the squares indicate HRs and the values under the squares indicate number of cases in each category. The size of the squares is proportional to the inverse variance of each effect size. BMI, body mass index; CI, confidence interval; HR, hazard ratio; MHN, metabolically healthy normal weight; MHO, metabolically healthy obesity; MHOW, metabolically healthy overweight; MUN, metabolically unhealthy normal weight; MUO, metabolically unhealthy obesity; MUOW, metabolically unhealthy overweight.

with an increased risk of MVE, with a little lower risk than that of stroke and IHD, but not with MCE. The observed HR for MVE in Chinese adults was much lower than that in U.S. women (1.39, 1.15–1.68) [9]. Interestingly, the Danish prospective Inter99 study in 6,238 men and women found that MHO was associated with an increased risk of IHD compared with MHN among men (3.1, 1.1–8.2) but not among women (1.8, 0.7–4.8) [29]. On the contrary,

**Table 3. Transition of BMI-metabolic health status from baseline to the second resurvey.**

| BMI-metabolic health status at baseline | BMI-metabolic health status at the second resurvey, number of participants (%) | | | | |
|---|---|---|---|---|---|
| | **MHN** | **MHOO** | **MUN** | **MUOO** | **Total** |
| MHN | 6,667 (67.53) | 1,498 (15.17) | 917 (9.29) | 790 (8.00) | 9,872 (100) |
| MHOO | 575 (9.83) | 2,820 (48.21) | 135 (2.31) | 2,320 (39.66) | 5,850 (100) |
| MUN | 271 (47.21) | 37 (6.45) | 173 (30.14) | 93 (16.20) | 574 (100) |
| MUOO | 134 (4.18) | 783 (24.40) | 125 (3.90) | 2,167 (67.53) | 3,209 (100) |

BMI, body mass index; MHN, metabolically healthy normal weight; MHOO, metabolically healthy overweight or obesity; MUN, metabolically unhealthy normal weight; MUOO, metabolically unhealthy overweight or obesity.

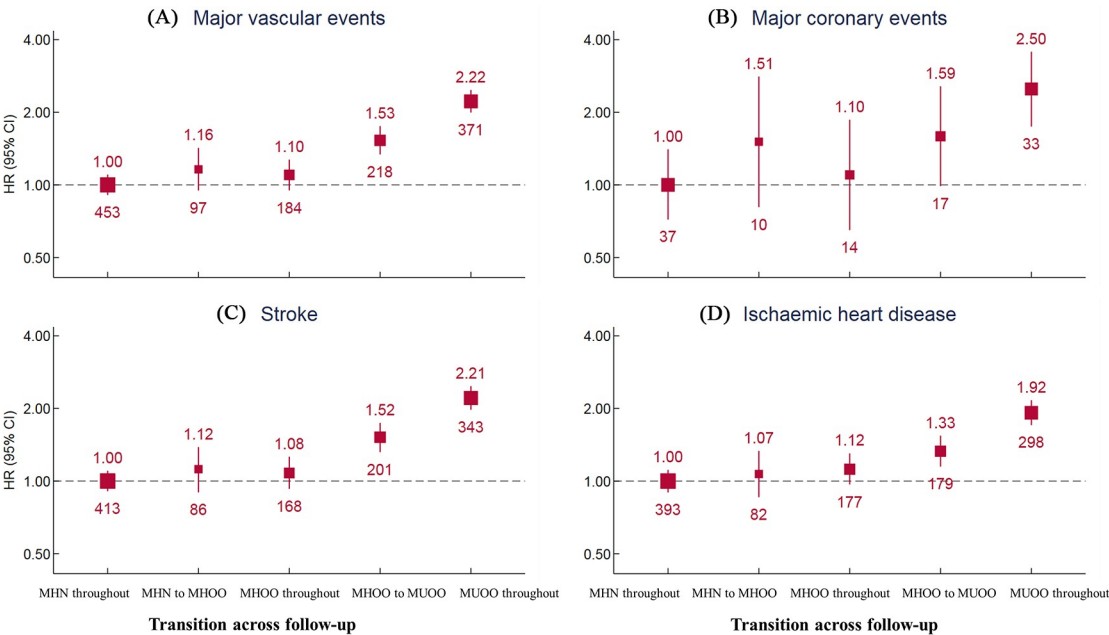

**Fig 4. Transition from metabolically healthy to unhealthy status and association with cardiovascular diseases subtypes risk.**
Values shown are the HR (95% CI) for **(A)** major vascular events, **(B)** major coronary events, **(C)** stroke, and **(D)** ischemic heart disease, by transition of BMI-metabolic health status. HRs are adjusted for age, study region, sex, education, household income, marital status, smoking, alcohol use, red meat intake, fresh fruits intake, fresh vegetables intake, physical activity, and family history of heart attack or stroke. Individuals with prior stroke, coronary heart disease, cancer are excluded from all analyses. BMI, body mass index; CI, confidence interval; HR, hazard ratio; MHN, metabolically healthy normal weight; MHOO, metabolically healthy overweight or obesity; MUOO, metabolically unhealthy overweight or obesity.

the present study with a large sample size (37,992 IHD cases) demonstrated that both MHO men and women showed increased risk of IHD in Chinese adults, with higher risk for women than for men. Such a discrepancy can be at least partly explained by the small sample size (323 participants developed IHD) in the Danish study. Our findings highlight that MHO is not a benign condition and is associated with an increased risk of CVD and that obesity remains a risk factor independent of common metabolic disorders among Chinese adults.

Our findings show that the CVD risk of metabolically unhealthy individuals was much higher than that of metabolically healthy individuals across BMI categories, which were consistent with the results of meta-analysis [17] and did not support previous findings that metabolic status was not more valuable than BMI in identifying individuals at risk for CVD [18, 30, 31]. Our results suggested that there existed considerable difference of CVD risk between MHO individuals and MUO individuals, though MHO was not a harmless condition for CVD. A potential mechanism underlying such difference is that, as compared with MUO individuals, MHO individuals have more adequate subcutaneous adipose tissue expansion, leading to more subcutaneous, less visceral fat mass, and lower ectopic fat deposition in the liver [6, 32–34], which have been associated with lower risks of metabolic abnormalities and CVD independent of BMI [35–37]. Our results also highlight that such an unhealthy metabolic obesity is associated with a substantial increase in vascular disease risk. Although maintenance of metabolic health may be difficult for individuals with obesity and overweight, it is a key target in preventing vascular disease. Our results suggest that recommendations for prevention vascular risk should highlight the importance of metabolic health maintenance across all BMI groups, including normal-weight individuals.

Furthermore, metabolic health status changes over time [9, 23, 38]. However, to our knowledge, our study is the largest Asian cohort study to investigate the transition of individuals with MHN over a longer follow-up. In the present study, we addressed the transition of metabolic health to unhealth status over 10 years of follow-up. A small number of participants (15.17%) changed from MHN to MHOO throughout the follow-up during follow-up. Almost 40% of the participants with initial MHOO converted to MUOO, similar with the conversion rates of 41%–48% over follow-up of 8–12 years in Western population [39–41]. Besides, in the Nurses' Health Study and the Whitehall II cohort study, over 80% and 50% of the participants converted from MHO to metabolic unhealthy status over 20 years [9, 39], suggesting a higher conversion rates with longer follow-up time.

High conversion rates from metabolically healthy to unhealthy obesity indicate that single point determination of metabolic health may be not sufficient to predict long-term vascular risk. By examining the transition of metabolic health status, 2 Western studies consistently showed that a large proportion of metabolically healthy subjects converted to an unhealthy status over time across all BMI categories, which was associated with an increased cardiovascular disease risk [9, 41]. In addition, the Danish prospective Inter99 study in a relatively small sample ($n$ = 6,238) showed metabolic changes were associated with a slightly higher risk for IHD [29]. However, it remains unclear whether such metabolic health changes over time affect the risk of subtypes of vascular disease such as stroke in Asians, especially in Chinese people. During the follow-up, we documented a large number of vascular cases and found that overweight or obese participates with stable healthy status had no significant association with vascular disease compared with participates with stable metabolic health normal weight. In contrast, metabolic health overweight or obese participants who changed to unhealthy status had a substantial higher risk of developing vascular disease such as IHD and stroke, which was much lower than overweight or obese participates with stable unhealthy status, indicating that a longer exposure to the metabolically unhealthy status is associated with a much higher vascular risk. Our results highlight that long-term maintenance of metabolic health is difficult for obese as well as overweight and normal-weight adults, but it is a key target in preventing vascular disease. Our study suggests that recommendations for vascular risk prevention should highlight the importance of maintaining metabolic health regardless of body weight, in addition to the current focus on the treatment of metabolic disorders. Furthermore, previous studies showed that MHO participants also had higher risk of diabetes [42], and both our study and previous studies further suggest that healthy overweight or obese people should also pay attention to maintaining metabolic health and monitoring glycemia, blood pressure, and lipid profile. Healthy lifestyle intervention should be enhanced among high-risk populations to prevent cardiometabolic disease.

The present study is thus far the largest cohort study on the associations of BMI-metabolic health status with CVD in Chinese adults. The strengths of our study include a prospective cohort design, relatively large number of cases, careful adjustment for established and potential risk factors for CVD, the repeated measurements, and documents of various CVD subtypes in Asian population. A further strength of the present study is that we considered the transition of metabolic health status during follow-up in estimating vascular risk; therefore, the accuracy of risk estimates might have been improved. However, several limitations of our study merit consideration. Firstly, there might be potential misclassification of metabolic status as we considered lipid-lowering drugs at baseline in contrast to studies using measured lipid components; however, the main analysis was replicated in the second resurvey involving a representative sample of baseline sample. In addition, information about lipid-lowering drugs was often considered to identify individuals with dyslipidemia [9, 43]. Secondly, metabolic health status was assessed using RPG, a good predictor of risk of CVD [44], instead of FPG,

which might have resulted in misclassification. However, we measured FPG for participants with RPG between 7.8 and 11.1 mmol/L. Thirdly, because of the low number of cases recorded in participants of the second resurvey, the transition effects of obesity from metabolic healthy to unhealthy status could not be evaluated with combination of overweight and obesity, and CIs for HRs in the second resurvey and the transition were large. Further large prospective cohort study is required to confirm the observations in the present study. It is worth noting that the results from the analysis in participants of the second resurvey were similar with those from baseline data, suggesting the robustness of our main findings. Besides, it is clear that participants changed from MHOO to MUOO and with stable MUOO had considerable higher risk of CVD, though the HR for stable MHOO is not precise enough to make the null conclusion. Fourthly, we had no direct measure of visceral fat, which is a better measure of obesity. Finally, although we adjusted for important confounders in the multivariable analysis, we cannot exclude the possibility of residual confounding factors due to unmeasured variables such as antihypertension drug use and level of HbA1c.

## Conclusion

In summary, our study shows that obesity, even without metabolic syndrome, is still an important risk factor for major vascular disease independent of these common metabolic disorders in Chinese adults. Our findings also support that recommendations for vascular prevention should highlight the importance of metabolic health maintenance across all BMI groups among Chinese adults. Importantly, our data suggest that metabolic health is a transient state for a large proportion of Chinese adults, with highest vascular risk for those who remain unhealthily obese. Closer attention should be paid to definition of metabolic health and its transition over time.

## Supporting information

**S1 Fig. Age-specific prevalence of MHO at baseline and second resurvey.** (A) Baseline; (B) the second resurvey. Prevalence by age was adjusted for sex and region. MHN, metabolically healthy normal weight; MHO, metabolically healthy obesity; MHOW, metabolically healthy overweight; MUN, metabolically unhealthy normal weight; MUO, metabolically unhealthy obesity; MUOW, metabolically unhealthy overweight.
(TIF)

**S2 Fig. Sex-specific prevalence of MHO at baseline and second resurvey.** (A) Baseline; (B) The second resurvey. Prevalence by sex was adjusted for age and region. MHN, metabolically healthy normal weight; MHO, metabolically healthy obesity; MHOW, metabolically healthy overweight; MUN, metabolically unhealthy normal weight; MUO, metabolically unhealthy obesity; MUOW, metabolically unhealthy overweight.
(TIF)

**S3 Fig. Adjusted HRs for types of cardiovascular disease by BMI-metabolic health status and baseline age.** Values shown are the HR (95% CI) for (A) major vascular events, (B) major coronary events, (C) stroke, and (D) ischemic heart disease, by BMI-metabolic health status and age. HRs are adjusted for study region, sex, education, household income, marital status, smoking, alcohol use, red meat intake, fresh fruits intake, fresh vegetables intake, physical activity, and family history of heart attack or stroke. The vertical lines indicate 95% CIs. The values above the squares indicate HRs and the values under the squares indicate number of cases in each category. The size of the squares is proportional to the inverse variance of each effect size. BMI, body mass index; CI, confidence interval; HR, hazard ratio; MHN,

metabolically healthy normal weight; MHO, metabolically healthy obesity; MHOW, metabolically healthy overweight; MUN, metabolically unhealthy normal weight; MUO, metabolically unhealthy obesity; MUOW, metabolically unhealthy overweight.
(TIF)

**S4 Fig. Test for nonlinear relationship between risk factors of metabolic health and major vascular events at baseline.** Results are adjusted for study region, sex, education, household income, marital status, smoking status, alcohol use, red meat intake, fresh fruits intake, fresh vegetables intake, physical activity, and family history of heart attack or stroke. $^*p < 0.05$, significant nonlinear relationship.
(TIF)

**S5 Fig. Adjusted HRs for major vascular events by the number of criteria of metabolic disorders participants met.** Values shown are the HR (95% CI) for major vascular events by the number of criteria of metabolic disorders participants met (A) at baseline, and (B) in the second resurvey. HRs are adjusted for study region, sex, education, household income, marital status, smoking, alcohol use, red meat intake, fresh fruits intake, fresh vegetables intake, physical activity, and family history of heart attack or stroke. The vertical lines indicate 95% CIs. The values above the squares indicate HRs and the values under the squares indicate number of cases in each category. $p$ for trend <0.05 at baseline and the second resurvey. CI, confidence interval; HR, hazard ratio.
(TIF)

**S1 Table. Characteristics of the study population in the second resurvey.**
(DOCX)

**S2 Table. Adjusted hazard ratios for vascular diseases by BMI-metabolic health status in the second resurvey.** BMI, body mass index.
(DOCX)

**S3 Table. Sensitivity analysis of association between BMI-metabolic health and types of cardiovascular disease.** BMI, body mass index.
(DOCX)

**S4 Table. Sensitivity analysis of association between changes of BMI-metabolic health and types of cardiovascular disease by using continuous variables.** BMI, body mass index.
(DOCX)

**S1 STROBE Checklist. Checklist of items that should be included in reports of cohort studies.** STROBE, Strengthening the Reporting of Observational Studies in Epidemiology.
(DOC)

**S1 Text. Analysis plan.**
(DOCX)

**S2 Text. Baseline and the second resurvey questionnaires.**
(PDF)

## Acknowledgments

The most important acknowledgment is to the participants in the study and the members of the survey teams in each of the 10 regional centers, as well as to the project development and management teams based at Beijing, Oxford, and the 10 regional centers. The members of the CKB collaborative group are as follows: **International Steering Committee:** Junshi Chen,

Zhengming Chen (PI), Robert Clarke, Rory Collins, Yu Guo, Liming Li (PI), Jun Lv, Richard Peto, Robin Walters. **International Coordinating Centre, Oxford:** Daniel Avery, Ruth Boxall, Derrick Bennett, Yumei Chang, Yiping Chen, Zhengming Chen, Robert Clarke, Huaidong Du, Simon Gilbert, Alex Hacker, Mike Hill, Michael Holmes, Andri Iona, Christiana Kartsonaki, Rene Kerosi, Ling Kong, Om Kurmi, Garry Lancaster, Sarah Lewington, Kuang Lin, John McDonnell, Iona Millwood, Qunhua Nie, Jayakrishnan Radhakrishnan, Paul Ryder, Sam Sansome, Dan Schmidt, Paul Sherliker, Rajani Sohoni, Becky Stevens, Iain Turnbull, Robin Walters, Jenny Wang, Lin Wang, Neil Wright, Ling Yang, Xiaoming Yang. **National Coordinating Centre, Beijing:** Zheng Bian, Yu Guo, Xiao Han, Can Hou, Jun Lv, Pei Pei, Chao Liu, Canqing Yu. **10 Regional Coordinating Centers: Qingdao CDC:** Zengchang Pang, Ruqin Gao, Shanpeng Li, Shaojie Wang, Yongmei Liu, Ranran Du, Yajing Zang, Liang Cheng, Xiaocao Tian, Hua Zhang, Yaoming Zhai, Feng Ning, Xiaohui Sun, Feifei Li. **Licang CDC:** Silu Lv, Junzheng Wang, Wei Hou. **Heilongjiang Provincial CDC:** Mingyuan Zeng, Ge Jiang, Xue Zhou. **Nangang CDC:** Liqiu Yang, Hui He, Bo Yu, Yanjie Li, Qinai Xu,Quan Kang, Ziyan Guo. **Hainan Provincial CDC:** Dan Wang, Ximin Hu, Jinyan Chen, Yan Fu, Zhenwang Fu, Xiaohuan Wang. **Meilan CDC:** Min Weng, Zhendong Guo, Shukuan Wu,Yilei Li, Huimei Li, Zhifang Fu. **Jiangsu Provincial CDC:** Ming Wu, Yonglin Zhou, Jinyi Zhou, Ran Tao, Jie Yang, Jian Su. **Suzhou CDC:** Fang liu, Jun Zhang, Yihe Hu, Yan Lu, Liangcai Ma, Aiyu Tang, Shuo Zhang, Jianrong Jin, Jingchao Liu. **Guangxi Provincial CDC:** Zhenzhu Tang, Naying Chen, Ying Huang. **Liuzhou CDC:** Mingqiang Li, Jinhuai Meng, Rong Pan, Qilian Jiang, Jian Lan,Yun Liu, Liuping Wei, Liyuan Zhou, Ningyu Chen Ping Wang, Fanwen Meng, Yulu Qin, Sisi Wang. **Sichuan Provincial CDC:** Xianping Wu, Ningmei Zhang, Xiaofang Chen,Weiwei Zhou. **Pengzhou CDC:** Guojin Luo, Jianguo Li, Xiaofang Chen, Xunfu Zhong, Jiaqiu Liu, Qiang Sun. **Gansu Provincial CDC:** Pengfei Ge, Xiaolan Ren, Caixia Dong. **Maiji CDC:** Hui Zhang, Enke Mao, Xiaoping Wang, Tao Wang, Xi zhang. **Henan Provincial CDC:** Ding Zhang, Gang Zhou, Shixian Feng, Liang Chang, Lei Fan. **Huixian CDC:** Yulian Gao, Tianyou He, Huarong Sun, Pan He, Chen Hu, Xukui Zhang, Huifang Wu, Pan He. **Zhejiang Provincial CDC:** Min Yu, Ruying Hu, Hao Wang. **Tongxiang CDC:** Yijian Qian, Chunmei Wang, Kaixu Xie, Lingli Chen, Yidan Zhang, Dongxia Pan, Qijun Gu. **Hunan Provincial CDC:** Yuelong Huang, Biyun Chen, Li Yin, Huilin Liu, Zhongxi Fu, Qiaohua Xu. **Liuyang CDC:** Xin Xu, Hao Zhang, Huajun Long, Xianzhi Li, Libo Zhang, Zhe Qiu.

## Author Contributions

**Conceptualization:** Canqing Yu, Yiping Chen, Lu Qi, Zhengming Chen, Tao Huang, Liming Li.

**Data curation:** Meng Gao, Jun Lv, Canqing Yu, Yu Guo, Zheng Bian, Ruotong Yang, Huaidong Du, Ling Yang, Zhengming Chen, Liming Li.

**Formal analysis:** Yu Guo, Ruotong Yang, Huaidong Du, Ling Yang, Yiping Chen, Zhongxiao Li, Xi Zhang, Junshi Chen, Tao Huang, Liming Li.

**Funding acquisition:** Liming Li.

**Investigation:** Zhengming Chen, Tao Huang, Liming Li.

**Methodology:** Meng Gao, Jun Lv, Zheng Bian, Ruotong Yang, Zhengming Chen.

**Writing – original draft:** Meng Gao, Yu Guo, Zheng Bian, Ruotong Yang, Huaidong Du, Ling Yang, Yiping Chen, Zhongxiao Li, Xi Zhang, Junshi Chen, Lu Qi, Zhengming Chen, Tao Huang, Liming Li.

**Writing – review & editing:** Lu Qi, Tao Huang, Liming Li.

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
