## [Editor Report · Decision Letter 0]

1 Mar 2020

Dear Dr Huang, 

Thank you for submitting your manuscript entitled "Metabolically healthy obesity and vascular, and non-vascular diseases among Chinese: a analysis based on the China Kadoorie Biobank." for consideration by PLOS Medicine.

Your manuscript has now been evaluated by the PLOS Medicine editorial staff and I am writing to let you know that we would like to send your submission out for external peer review.

Kind regards,

Helen Howard, for Clare Stone PhD 

Acting Editor-in-Chief

PLOS Medicine 

plosmedicine.org

---

## [Decision Letter · Decision Letter 1]

30 Mar 2020

Dear Dr. Huang,

Thank you very much for submitting your manuscript "Metabolically healthy obesity and vascular, and non-vascular diseases among Chinese: a analysis based on the China Kadoorie Biobank." (PMEDICINE-D-20-00639R1) for consideration at PLOS Medicine. 

[LINK]

In light of these reviews, I am afraid that we will not be able to accept the manuscript for publication in the journal in its current form, but we would like to consider a revised version that addresses the reviewers' and editors' comments. Obviously we cannot make any decision about publication until we have seen the revised manuscript and your response, and we plan to seek re-review by one or more of the reviewers. 

We expect to receive your revised manuscript by Apr 20 2020 11:59PM. Please email us (plosmedicine@plos.org) if you have any questions or concerns.

We look forward to receiving your revised manuscript. 

Sincerely,

Clare Stone

Chief Editor 

PLOS Medicine

plosmedicine.org

Title: Please revise your title according to PLOS Medicine's style. Your title must be nondeclarative and not a question. It should begin with main concept if possible. "Effect of" should be used only if causality can be inferred, i.e., for an RCT. Please place the study design ("A randomized controlled trial," "A retrospective study," "A modelling study," etc.) in the subtitle (ie, after a colon). Maybe the study design should be a large cohort analysis using Chinese Kadoorie Biobank data.

Data – you say some restrictions will apply, then say all data is available in the MS and Supp Files. Please state what data is available and what is not and why. 

Abstract – this should be formatted with 3 sections: Background, Method and Findings, Conclusion. Please format accordingly. 

Abstract – use months as well as years for recruitment and tell us the cities / regions where recruitment took place, if known; Please use 95% Cis as well as p values for all quantifiable data (both here and throughout the manuscript); Please ensure summary demographic information is provided, including gender ratio and mean age and number of smokers, etc; Please provide – as the last sentence of the ‘Methods and Findings’ section a sentence on the limitations of the study; 

Refs in main text, these need to be presented in square brackets and not using superscript. 

In the main text where you use 95%Cis, please also add p values. 

Please include line numbers on your revised version.

Please be careful of deriving cause where non can be drawn as this is not a trial: “Our data provide evidence…” and “In summary, our study provides evidence that obesity, even without metabolic syndrome…”

Please use the "Vancouver" style for reference formatting, and see our website for other reference guidelines https://journals.plos.org/plosmedicine/s/submission-guidelines#loc-references (noting remove ital font specifically)

Did your study have a prospective protocol or analysis plan? Please state this (either way) early in the Methods section.

Please ensure that the study is reported according to the STROBE guideline, and include the completed STROBE checklist as Supporting Information. Please add the following statement, or similar, to the Methods: "This study is reported as per the Strengthening the Reporting of Observational Studies in Epidemiology (STROBE) guideline (S1 Checklist)."

Comments from the reviewers:

Reviewer #1: I confine my remarks to statistical aspects of this paper. Unfortunately, I have some major issues to resolve before I can recommend publication.

The main issue is that the authors categorize all the independent variables. This is a mistake. In *Regression Modeling Strategies* Frank Harrell lists 11 problems with this and sums up "Nothing could be more disastrous". I wrote a blog post that shows some of the problems, graphically: https://medium.com/@peterflom/what-happens-when-we-categorize-an-independent-variable-in-regression-77d4c5862b6c

Instead, BMI, waist circumference and so on should be left continuous and splines used to look for nonlinearities. In addition, metabolic health status should be a continuous variable based on a combination of the factors listed on p. 6. One way to get such a measure would be factor analysis. (Categories here *might* be useful for transitions, although I think continuous is better there, too; but for the regressions, continuous IVs are definitely better).

Also, BMI is a poor measure of adiposity, even if you don't categorize. Waist hip ratio is much better. BMI has the advantage of being easier to measure, but since you already measured waist, why did you ignore hips? Waist hip ratio is a good measure of adiposity.

More specific comments (NOTE: Line numbers would have made this easier)

p. 5 Why only measure waist size? Why not adjust for height? (I know there are some studies using these cutoffs, but surely a 90 cm waist on a 2 meter tall man is not the same as one on a 1.7 m tall man. 

 Why were these glucose levels used?

p. 7 Near the bottom, what does "if appropriate" mean?

p. 8 As noted, none of the continuous IVs should be categorized (food intake, income, exercise)

Figure 1 might be better as a mosaic plot

Figure 4 for this, categories might be useful. If you use categories, figure 4 should be a table with initial status and final status crossed and frequencies in the cells. A visual representation could be a mosaic plot. If you use continuous measures, you could make a scatterplot with a loess or other smoother.

Peter Flom

Reviewer #2: The authors investigated association of metabolic health (MH) across BMI groups and of changes in MH status with the risk of CVD event, based on a large cohort study in China. MH obesity was related to an increased risk, as was a change from MH to an unhealthy metabolic status during follow-up.

Major comments:

1. The study is largely confirmatory, even if larger-scale cohort studies are so far lacking from China. That MHO defined by the absence of the metabolic syndrome is still related to an increased risk have been shown by meta-analyses of several cohort studies. Also, it is well described that an unhealthy phenotype strongly increases risk across all BMI categories. Furthermore, that changes to an unhealthy phenotype increases risk has also been described before. Thus, the study lack novelty overall.

2. While the cohort is indeed large, the resurveys to update BMI and MH status only included a minor fraction of the original cohort. As a result, the analysis considering MH status at the 2nd resurvey as baseline is severely underpowered. With the very short follow-up (2-3 years) and low number of cases this analysis isn't informative (easily observable by the large CIs for HRs)

3. This problem also affects the analysis on MH changes. First, overweight and obese individuals were combined in this analysis, which limits interpretation. Furthermore, while transition to MU in overweight/obese can be evaluated, such a transition in normal-weight could not. It also remains unclear how a stable MHO status is related to risk. The HR for MHOO (1.10) is not significant, however, the effect size isn't materially different from the analysis using baseline MHO (1.08). Still, the lack of precision doesn't allow to make conclusions. 

4. From the study methods it remains unclear how complete assessment of case status in the cohort is. How complete is the population coverage of disease registries? What is the proportion of participants with active follow-up and how were self-reports verified?

5. The discussion on p. 22 highlights that it is a novel or different finding that individuals with MU have higher risk across all BMI groups. This is a false statement as many studies have found a similar picture before, as has been summarized in meta-analysis (Ref. 16) or described from large scale cohorts (e.g. Lasalle 2018 - one of the largest studies, other Refs e.g. 8)

6. The authors use one definition, the metabolic syndrome, although this definition has been discussed to be not strict enough to detect a true low-risk group (risk factors can still be present). Particularly the large sample from the baseline cohort could be used to evaluate alternative definitions of MH, e.g. the absence of any metabolic syndrome component. 

Reviewer #3: Gao et. al report on a large observational study including data from 512,715 adults included in the China Kadoorie Biobank. They examine a controversial exposure of metabolically healthy obesity and its association with cardiovascular disease. This is an important study extending known data in a large population from China.

The paper could be strengthened with the following revisions:

Title: 0.5 million seems odd since it is less than 1; would exclude the number from the title

Abstract: 

1. I am unclear after reading the methods what the BMI categories were, how MHO and MHN was defined, how changes in metabolic health status were defined, and what statistical analysis was done to examine risk

Introduction:

1. remove phrasing that labels people as "obese" rather should state "people with obesity"

2. MHO is itself a misnomer implying that obesity can be healthy. If this is warranted, the introduction could state this or can be in the discussion

3. Instead of body shape, would consider rephrasing to say "adiposity distribution" 

4. Both in the abstract and in the introduction, the following words are used interchangeably: CVD, vascular risk, MVE, MCE, cardiovascular events. I think you should pick one and be consistent throughout unless they represent different things that are not well defined and are confusing. The most traditional one to use would be to consistently refer to all events as cardiovascular disease (CVD) and refer to each subtype as indicated.

5. It would be nice to reference the recent BRICS study in Circulation about changing rates of CVD mortality in China and other low and middle-income countries and how obesity/diabetes may play a role in the first paragraph.

Methods:

1. For the reader, please define and provide a reference as to why you chose the BMI cutoffs that you did. these may be appropriate for individuals of Asian ancestry, but they are not universal.

Discussion:

1. Instead of 0.5 million just state the sample size or approximately 500,000

2. Could be shortened, specifically the paragraph on page 12 that extends to page 13 is too long. 

3. Paragraph that begins on page 14 should be rephrased to say "our findings show.." not the past tense

4. The claim on page 14, that no other Asian cohort study has investigated the transition of individuals with MHO is not true. in the Multi-Ethnic Study of Atherosclerosis, patients of Chinese ancestry were included and they have studied this. Would revise and include this. However, the current study has a much larger sample size.

5. The paragraph on page 15 is also far too long and could be cut in half.

6. Limitations should include the fact that no direct measure of visceral adiposity of %body fat was included and the discussion around how we know that BMI is a poor measure of obesity

7. The limitation of salt intake seems a bit odd since we are not talking about blood pressure anywhere. I would include the limitation of not having measured HbA1c which is the clinically relevant way to measure presence of pre-diabetes and diabetes

The tables and figures are very nicely done and display the results well.

[LINK]

---

## [Decision Letter · Decision Letter 2]

18 Jun 2020

Dear Dr. Huang,

Thank you very much for submitting your manuscript "Metabolically healthy obesity, transition to unhealthy metabolic status and vascular disease in 0.5 million Chinese adults: a prospective cohort study" (PMEDICINE-D-20-00639R2) for consideration at PLOS Medicine. 

As you will see, one of the referees continued to raise concerns about your study. As such, we reached out to our Academic Editor for further thoughts. s/he has added comments and we ask you to revise according also to these also. They are pasted below. 

[LINK]

In light of these reviews, I am afraid that we will not be able to accept the manuscript for publication in the journal in its current form, but we would like to consider a revised version that addresses the reviewers' and editors' comments. Obviously we cannot make any decision about publication until we have seen the revised manuscript and your response, and we plan to seek re-review by one or more of the reviewers. 

We expect to receive your revised manuscript by Jul 09 2020 11:59PM. Please email us (plosmedicine@plos.org) if you have any questions or concerns.

We look forward to receiving your revised manuscript. 

Sincerely,

Clare Stone, PhD

Acting Chief Editor 

PLOS Medicine

plosmedicine.org

please address points below. 

Academic Editor's points:

This study is very interesting, investigating the association of transition state from healthy overweight or obesity to unhealthy overweight or obesity with CVDs. If an overweight or obese participant had no metabolic disorders, the risks of CVDs would not increase. But, if an overweight or obese participant accompanied with metabolic disorders, the risks of CVDs would increase. The results suggested overweight or obese persons should not only pay attention to their weight or BMI, but also need to check the metabolic indexes regularly. 

Several questions or suggestions were as follows. 

1. The sample was around 0.5 million for the baseline. However, the sample of resurvey was only close to 20 thousand. It was said that this represented a 5% random sampling from the baseline sample. So, please clarify the representativeness of the resurvey sample with the baseline sample further. Thus, the title of this paper might be inappropriate. Should it be modified? 

2. Due to the small number of resurvey sample, the numbers of follow-up events of different vascular diseases in different categories of healthy or unhealthy overweight or obese participants were few. So, the grouping of overweight and obesity into one category when assessing the transition to healthy or unhealthy metabolic obesity was suggested. 

3. Please further strengthen that even healthy overweight or obesity state could also result in diabetes and other related metabolic disorders in discussion. So, even healthy overweight of obesity should also be paid attention to, to monitor glyceamia, blood pressure, and lipid profile, etc. And, healthy lifestyle intervention should be enhanced. Because the abnormalities of those indexes would indicate increasing CVDs risks.

Comments from the reviewers:

Reviewer #1: The authors have addressed my concerns and I now recommend publication

Peter Flom

Reviewer #2: I cannot see where the authors have indeed addressed previous comments. The main limitations of this analysis remain as they had been:

- The study findings are largely confirmatory - I am not convinced that providing regional data from China is creating a substantial novelty. Why would previous studies on MH and transitions not be largely generalizable to Chinese populations? 

- The sample with follow-up information on MH status and obesity is small. The combination of overweight and obese into one group doesn't allow to evaluate transition effects from MH in obesity.

- Effect estimates are largely imprecise for analyses using the second survey as baseline for the most relevant group (MHO). The argument of the authors that these data are used to "validate" results from the baseline assessment isn't convincing - this is only possible for the least interetsing group (it is well established that individuals with obesity plius multiple cardiovascular risk factors are at higher risk compared to normal-weight health inmdividuals!)

- I also don't agree with the authors notion, that it is not well established that MUH is related to higher risk across different BMI categories. While it is certainly easy to pick individual studies which might not have shown this, meta-analyses of cohort stdueis using the metabolic syndrome don't support this argument.

- It is unfortunate that the autors did not attempt to evaluate alternative definitions of MH/MUH merely based on the argument that absence of any risk factor was extremly rare among obese. Many MH definitions used previously exclude WC (given the high correlation with BMI this measure isn't very informative to subgroup obese) for example. 

Reviewer #3: Gao et. al have revised their manuscript examining the association of obesity (+/- metabolic healthy) and major cardiovascular events overall and by subtype in 500,000 Chinese adults across 10 regions (urban and rural). This is a very nicely done study and the authors have been very responsive to all suggestions--the abstract is clear and succinct and the introduction and discussion have been markedly improved. I have no additional major suggestions at this time and believe that this is a very timely much-needed addition.

In terms of the tables and figures, I find Table 1 a bit overwhelming and wonder if the baseline survey data could be presented here and the re-survey moved to the supplement and similarly for Table 2.

[LINK]

---

## [Editor Report · Decision Letter 3]

11 Aug 2020

Dear Dr. Huang,

Thank you very much for re-submitting your manuscript "Metabolically healthy obesity, transition to unhealthy metabolic status and vascular disease in Chinese adults: a prospective cohort study" (PMEDICINE-D-20-00639R3) for review by PLOS Medicine.

I have discussed the paper with my colleagues and the academic editor and it was also seen again by reviewers. I am pleased to say that provided the remaining editorial and production issues are dealt with we are planning to accept the paper for publication in the journal.

[LINK]

We look forward to receiving the revised manuscript by Aug 18 2020 11:59PM. 

Sincerely,

Clare Stone, PhD

Acting Chief Editor 

PLOS Medicine

plosmedicine.org

Requests from Editors:

The author summary needs to be in bulleted form and with 3 headings. Please do look at other published Research articles for guidance. It should also be written for the non-specialist and not repeat the abstract. 

In the main text, please include a space between the last letter of the word and the following square bracket for refs – For example, instead of 17 million deaths annually[1], it should read 17 million deaths annually [1]. Correct throughout, please.

Please ensure all questionnaires are provided as Supplementary Files and translated where necessary. 

At line 334, Please mention their Chinese population and avoid "a little lower risk" as this is too vague. Say what the risk is. 

Please remove the word "prospective" from the title (I believe that this is a retrospective analysis of a prospectively gathered dataset).

Comments from Reviewers:

[LINK]

---

## [Editor Report · Decision Letter 4]

11 Sep 2020

Dear Dr Huang, 

On behalf of my colleagues and the academic editor, Dr. Weiping Jia, I am delighted to inform you that your manuscript entitled "Metabolically healthy obesity, transition to unhealthy metabolic status and vascular disease in Chinese adults: a cohort study" (PMEDICINE-D-20-00639R4) has been accepted for publication in PLOS Medicine. 

PRODUCTION PROCESS

PRESS

PROFILE INFORMATION

Thank you again for submitting the manuscript to PLOS Medicine. We look forward to publishing it. 

Best wishes, 

Clare Stone, PhD

Managing Editor 

PLOS Medicine

plosmedicine.org